# Transcriptome Analysis Reveals the AhR, Smad2/3, and HIF-1α Pathways as the Mechanism of Ochratoxin A Toxicity in Kidney Cells

**DOI:** 10.3390/toxins13030190

**Published:** 2021-03-06

**Authors:** Min Cheol Pyo, In-Geol Choi, Kwang-Won Lee

**Affiliations:** Department of Biotechnology, College of Life Science & Biotechnology, Korea University, Seoul 02841, Korea; reve03@korea.ac.kr (M.C.P.); igchoi@korea.ac.kr (I.-G.C.)

**Keywords:** ochratoxin A, aryl hydrocarbon receptor, Smad2/3, hypoxia-inducible factor-1α, RNA-sequencing

## Abstract

Ochratoxin A (OTA) is a mycotoxin occurring in foods consumed by humans. Recently, there has been growing global concern regarding OTA toxicity. The main target organ of OTA is the kidney, but the mechanism underlying renal toxicity is not well known. In this study, human-derived proximal tubular epithelial cells, HK-2 cells, were used for RNA-sequencing (RNA-seq) and transcriptome analysis. In total, 3193 differentially expressed genes were identified upon treatment with 200 nM OTA in HK-2 cells; of these, 2224 were upregulated and 969 were downregulated. Transcriptome analysis revealed that OTA significantly affects hypoxia, epithelial-mesenchymal transition (EMT), apoptosis, and xenobiotic metabolism pathways in kidney cells. Quantitative real-time PCR analysis showed gene expression patterns similar to RNA-seq analysis. Expression of EMT markers (E-cadherin and fibronectin), apoptosis markers (caspase-3 and Bax), and kidney injury molecule-1 (KIM-1) was suppressed by inhibiting AhR expression using siRNA, and the related transcription factors, Smad2/3, and HIF-1α were downregulated. Smad2/3 suppression with siRNA could inhibit fibronetcin, caspase-3, Bax, and KIM-1 expression. Fibronetcin, caspase-3, Bax, and KIM-1 expression could be increased with HIF-1α suppression with siRNA. Taken together, these findings suggest that OTA-mediated kidney toxicity via the AhR-Smad2/3-HIF-1α signaling pathways leads to induction of EMT, apoptosis, and kidney injury.

## 1. Introduction

Ochratoxin A (OTA) is a mycotoxin that occurs naturally in fungi such as *Aspergillus spp.* and *Penicillium spp.* OTA is known to exist in a variety of food groups such as cereals, cocoa, coffee, nuts, milk, beer, and wine, which are frequently consumed in daily life [1,2]. OTA (C_20_H_18_O_6_NCl, molecular weight: 403.8) is a colorless, odorless, slightly water-soluble crystalline compound belonging to the para-chlorophenolic group that structurally contains a dihydroisocoumarin moiety linked by an amide bond to L-phenylalanine [3]. It is more stable to heat than other mycotoxins and is absorbed into the body through the gastrointestinal tract with food consumption. Ingested OTA is rapidly absorbed through the jejunum with a bioavailability of 97% [4]. OTA binds strongly to albumin in serum and has a very long half-life, eventually accumulating in the body [5]. OTA is one of the most harmful fungal toxins and is classified in Group 2B as a possible human carcinogen by the International Agency for Research on Cancer [6]. The nephrotoxicity, hepatotoxicity, immunotoxicity, neurotoxicity, and genotoxicity of OTA have been reported in various animal species [7,8,9].

OTA has various possible targets such as the liver, immune system, heart and brain; however, owing to the function of the organ, its main target is the kidneys [10,11,12]. The kidneys balance the metabolism of salt and water, regulate the levels of electrolytes in blood, and play several important physiological roles including maintaining the stable composition of blood and secretion of hormones [13,14]. The kidneys also receive about 25% cardiac output and function to filter the equivalent volume of extracellular fluid from plasma. Further, the concentration of toxic substances is increased as water and solutes are reabsorbed in the process of removing waste from systemic circulation and excretion to urine. As a result, the risk of renal exposure to toxic substances is increased [15]. Due to the high renal blood flow per tissue weight, a relatively large amount of OTA is delivered to the kidneys compared to other organs [16]. Further, OTA accumulation in kidney tissues, especially through reabsorption, results in the highest concentration of OTA to be detected throughout the body [17].

Transcriptome sequencing (RNA-seq) technology has been used to evaluate and analyze the overall response caused by toxic substances or stressors [18,19]. Although there is a limitation of RNA-seq that not all RNAs have been used to produce proteins, the use of RNA-seq technology to analyze differentially expressed genes (DEGs) is a reliable method for understanding the interactions between specific molecules and for conducting research on the mechanisms that cause toxicity [20,21]. DEG analysis has become a valuable tool for studying molecular mechanisms in response to toxic substances including mycotoxins [22]. Transcriptome analysis studies on OTA have been performed in a variety of cell lines and mammalian model systems [23,24,25]. Therefore, much information about the mechanism of OTA toxicity in the kidney can be obtained through transcriptional profiles obtained using RNA-sequencing.

Although much is known about the renal toxicity of OTA such as inflammation, apoptosis and pyroptosis [26,27], studies on the mechanisms underlying renal toxicity are still insufficient. Therefore, in the present study, transcriptome analysis was used to estimate changes in renal damage in response of OTA. In this research, OTA-induced renal toxicity such as epithelial-mesenchymal transformation (EMT), apoptosis and hypoxia were investigated in human proximal tubular epithelial HK-2 cells.

## 2. Results

### 2.1. Analysis of DEGs

To elucidate the molecular mechanisms responsible for OTA toxicity to the kidneys, transcriptome sequencing and analysis were conducted in HK-2 cells. Differential gene expression analysis of HK-2 cells following OTA treatment resulted in a total of 3193 DEGs, of which 2224 were upregulated and 969 were downregulated (Figure 1). As shown in the scatter plot, the expression level of certain genes was significantly (*p* < 0.05) different between the control group and the OTA-treated group, suggesting that the gene expression of HK-2 cells was affected by OTA (Figure 2).

### 2.2. Functional Annotation of DEGs

To further explain the molecular function of DEGs identified in HK-2 cells exposed to OTA, GO and KEGG enrichment analysis were performed. As shown in Figure 3A–C, in case of GO analysis for the upregulated and downregulated DEGs, there are three categories, biological process, molecular function, and cellular component, and the top 15 terms for each category are shown based on *p*-values < 0.05 (Figure 3D). Representative subsets in the biological process include cell–cell adhesion, rRNA processing, and viral process; molecular functions include protein binding, poly(A) RNA binding, and cadherin binding involved in cell–cell adhesion. Cellular components include nucleoplasm, membrane, and cytosol. KEGG pathway analysis also showed the top 15 pathways. As shown in Figure 3D, protein processing in the endoplasmic reticulum, insulin resistance, thyroid hormone signaling pathway, proteoglycan in cancer, renal cell carcinoma, adherent junction, and others were found. Among the various KEGG pathways, renal cell carcinoma was the pathway with the highest fold-enrichment value. This is also associated with hypoxia and EMT [28,29].

### 2.3. Gene Set Enrichment Analysis of Genes Related to OTA

To identify the potential functions of DEGs, GSEA was performed using the MSigDB hallmark gene set. As shown in Table 1, 20 gene sets regulated by OTA in HK-2 cells were identified. Among them, we focused on hypoxia, EMT, apoptosis, and xenobiotic metabolism as pathways for the toxic mechanism of OTA based on the results of KEGG analysis.

### 2.4. Validation of Gene Expression Patterns Using qRT-PCR

To validate the transcriptome results of DEGs, qRT-PCR analysis was performed, and a total of 11 genes were selected based on the pathway results in the Hallmark gene sets through GSEA. Gene expression levels from qRT-PCR analysis showed similar trends and magnitudes of changes compared to the transcriptome analysis (Figure 4). Correlation analysis of the 11 DEGs yielded an *R*^2^ of 0.8097, demonstrating excellent linearity between the RNA-seq and qRT-PCR results, confirming the reliability of the RNA-seq data.

### 2.5. Effects of AhR Knockdown on EMT and Kidney Injury-Related Marker Expression

We have reported that AhR regulated phase I and II reactions leading to OTA-induced hepatotoxicity [30]. After knockdown of AhR expression in HK-2 kidney cells using siRNA and OTA treatment, mRNA and protein expression was confirmed by qRT-PCR and Western blot analysis, respectively, for markers of EMT, apoptosis, and kidney injury. As shown in Figure 5, E-cad mRNA and protein expression was increased in cells with AhR knockdown. RNA-seq and qPCR data showed that E-cad expression was decreased by OTA treatment in HK-2 cells (Figure 4A). In contrast, FN expression was decreased by the siAhR transfection. The mRNA and protein expression of caspase-3 and Bax, which are pro-apoptotic markers, was suppressed by AhR knockdown. The kidney injury marker, KIM-1, was also suppressed in siAhR-transfected cells.

### 2.6. Effect of AhR Knockdown on Smad2/3 and HIF-1α Expression

qRT-PCR and Western blot were performed to identify the transcription factors related to EMT and hypoxia upon AhR knockdown in HK-2 cells and treatment with OTA. The expression of Smad2/3, an EMT-related transcription factor, and that of HIF-1α, a hypoxia-related transcription factor, were also suppressed upon OTA treatment of siAhR-transfected cells (Figure 6).

### 2.7. Effects of Smad2/3 and HIF-1α Knockdown on EMT and Renal Injury-Related Markers Expression

To confirm the association between transcription factors Smad2/3 and HIF-1α and the processes of EMT, apoptosis, and kidney injury caused by OTA treatment, Smad2/3 and HIF-1α were knocked down with siRNA in HK-2 cells, followed by OTA treatment. The expression of EMT, apoptosis, and kidney injury-related markers was confirmed by qRT-PCR and Western blot. E-cad, which showed decreased expression in OTA treatment, was increased with siSmad2/3 knockdown, whereas siSmad2/3 transfection decreased the expression of FN, caspase-3, Bax, and KIM-1, which were increased by OTA (Figure 7A,B). On the contrary, when HIF-1α expression was knocked down, E-cad expression was decreased further, and the expression of FN, caspase-3, Bax, and KIM-1 was increased at the mRNA and protein levels (Figure 7).

## 3. Discussion

OTA is a mycotoxin found in various foods that are widely ingested by humans and, owing to its long half-life and thermal stability in food, is highly toxic to humans and animals. It is absorbed through the small intestine after oral intake of OTA [31,32]. In different animal species and in various organs of such animals, OTA toxicity was found [30,33,34,35]. Research on the mechanism by which OTA causes toxicity, however, is still limited. In this study, by transcriptome analysis using RNA sequencing technique, we attempted to analyze the OTA-regulated pathways and studied the relationship between the pathways and the toxicity of OTA. Transcriptome research, by deciphering the structure and role of the genome, is a tool for studying the pathological processes of disease and offers valuable knowledge on the range of response to pathogens and environmental stress [36]. We focused on hypoxia, xenobiotic metabolism, EMT, and apoptosis, along with the processes that occur in OTA-treated kidney cells, based on previous studies of OTA-induced toxicity pathways [30,37,38] and our transcriptome outcome (Table 1).

Xenobiotic metabolism refers to the metabolic or detoxification processes occurring in the liver and kidneys when toxic substances are biotransformed in the body [39]. Aflatoxin A or benzo(a)pyrene are activated by xenobiotic metabolism [40,41]. During xenobiotic metabolism, a transcription factor called AhR, which regulates the expression of cytochrome P450 (CYP) enzymes is activated. Subsequent activation of AhR might produce oxidative stress and ROS, resulting in cancer induction and immune function suppression [42,43]. Our previous in vitro and in vivo study using HK-2 cells and ICR male mice reported significantly increased mRNA and protein expression of CYP1A1 and CYP1A2 in the group treated with OTA compared to the control group, and when HK-2 cells were exposed to OTA, AhR-activated renal damage was caused by excessive ROS production during the metabolic processes involving CYP enzymes [35]. Generation of excessive oxidative stress also causes apoptosis and EMT [44,45].

Apoptosis occurs through two pathways, the intrinsic and extrinsic apoptotic pathways, both of which induce the caspase cascade [46]. Apoptosis is considered an important process in cell cell death [47]. Several studies have documented the induction of cellular apoptosis via oxidative stress upon OTA exposure [48,49]. Based on our RNA-seq and qRT-PCR results in HK-2 cells, the expression of Bax and caspase-3 was reduced after knocking down the expression of AhR indicating that OTA-induced apoptosis is triggered via AhR (Figure 5).

EMT is an important event in tumor metastasis where epithelial cells lose their epithelial function and cell–cell adhesion to acquire the mobility and invasive features of mesenchymal cells [50]. This transformation leads to loss of cell junctions and separation between cells and the basement membrane, resulting in cancer stem cell-like properties [51]. Further, instability of adherens junction due to downregulation of E-cadherin in epithelial cells and upregulation of fibronectin in mesenchymal cells are the representative features of EMT [52]. Many studies have shown that EMT progresses by the activation of AhR [53,54]. In this experiment, downregulation of E-cad and upregulation of FN with OTA treatment were confirmed through RNA-seq and qRT-PCR, and OTA was found to induce EMT (Figure 4). Upon OTA treatment after suppressing AhR expression, Smad2/3, an EMT-induced transcription factor, was suppressed (Figure 6) with E-cad upregulation and FN downregulation, which are regulated by Smad2/3 (Figure 5). These results indicate that OTA induces EMT through the AhR-Smad2/3 pathway. Further, studies have shown that apoptosis is induced in the liver and kidney through the Smad3 pathway [55,56]. Smad2/3 expression was suppressed to validate the association between EMT and apoptosis, and caspase-3, Bax, and KIM-1 expressions were decreased on OTA treatment. These results showed that like EMT, OTA induces apoptosis and kidney injury through the AhR-Smad2/3 pathway (Figure 7).

Hypoxia is a condition in which the oxygen supply in cells and organs is insufficient. It is expected to occur in kidney tissues due to microcirculation disruption and hypoperfusion [57]. In various clinical and laboratory settings, hypoxia is also reported to occur in both acute kidney injury and chronic kidney disease [58,59]. The role of HIF-1α in hypoxia-induced apoptosis is controversial, but some studies have shown that under particular conditions, HIF-1α protects cells against apoptosis [60,61,62]. It is also known that HIF-1α protects cells against hypoxic kidney damage by upregulating cell-protective factors such as VEGF, HO-1, and erythropoietin [63,64,65]. Moreover, studies in rat models of renal ischemia/reperfusion have shown that chemical damage is increased when HIF-1α is inhibited, while accumulation of HIF-1α has a protective effect against damage [66,67]. It is known that EMT and fibrosis progress after renal ischemia-reperfusion injury [57] [57,68]. HIF-1α plays a protective role against IRI in the kidney [67]. It protects against IRI by decreasing renal IRI-induced expression of fibrosis and alpha-SMA through increasing HIF-α alpha levels [69]. In this study, the HIF-1α expression in HK-2 cells was knocked down using siRNA. The expression of Bax, caspase-3, KIM-1, and FN was found to be increased, whereas E-cad decreased. These results show that HIF-1α is stabilized due to an adaptive response to OTA-induced hypoxia and serves to suppress the progression of apoptosis, EMT, and kidney injury (Figure 7).

In conclusion, EMT and apoptosis in the kidney are triggered by OTA and effectively lead to kidney injury. In addition, EMT, apoptosis, and kidney damage caused by OTA can occur in association with the AhR-Smad2/3-HIF-1α pathways. Thus, our findings can contribute to understanding the mechanism and prevention of renal toxicity of OTA, such as OTA-induced EMT and apoptosis.

## 4. Materials and Methods

### 4.1. Chemicals

Fetal bovine serum (FBS) and Roswell Park Memorial Institute (RPMI) 1640 medium were purchased from Gibco (Grand Island, NY, USA). Penicillin-streptomycin and trypsin-ethylenediaminetetraacetic acid (EDTA) were purchased from Hyclone (Logan, UT, USA). Dextrose, sodium bicarbonate, 4-(2-Hydroxyethyl) piperazine-1-ethanesulfonic acid (HEPES), dimethyl sulfoxide (DMSO), and thiazolyl blue tetrazolium bromide (MTT) were purchased from Sigma-Aldrich (St. Louis, MO, USA). Bicinchoninic acid (BCA) kit and SuperSignal^TM^ west Femto chemiluminescent substrate were purchased from Thermo Fisher Scientific (Waltham, MA, USA). Antibodies against Smad2/3 and kidney injury molecule-1 (KIM-1) were purchased from Cell Signaling (Denver, MA, USA). Glyceraldehyde-3-phosphate dehydrogenase (GAPDH) antibody was obtained from Millipore (Temecula, CA, USA). Antibodies against AhR, hypoxia-inducible factor 1-alpha (HIF-1α), epithelial cadherin (E-cad), fibronectin (FN), and Bcl-2-associated X protein (Bax), caspase-3 were purchased from Santa Cruz Biotechnology (Dallas, TX, USA). AhR siRNA (5′-GUGACUUGUACAGCAUAAUTT-3′) was purchased from GenePharma (Shanghai, China), Smad2/3 siRNA (sc-37238) was purchased from Santacruz Biotechnology (Santacruz, CA, USA), and HIF-1α siRNA (SASI_HS02_00332063) was purchased Sigma-Aldrich (St. Louis, MO, USA).

### 4.2. Cell Culture

HK-2, a human-derived proximal tubule epithelial cell line, was purchased from Korea Cell Line Bank (Seoul, Korea). HK-2 cells were cultured in RPMI 1640 medium containing 0.11 g/L sodium pyruvate, 2.5 g/L dextrose, 2.383 g/L HEPES, 100 U/mL penicillin, and streptomycin, 2 g/L sodium bicarbonate, and 10% FBS (*v/v*), in an incubator at 37 °C with 5% CO_2_. Cells were cultured until the cell density reached 80–90%.

### 4.3. OTA Treatment of HK-2 Cells

OTA (purity: 99%, Cfm Oskar Tropitzsch GmbH, Marktredwitz, Germany) was dissolved in DMSO at a concentration of 200 μM and stored at −20 °C until used in experiments. In the experiments, to prepare 200 nM OTA, it was diluted into RPMI1640 media. In both the OTA treated group and the non-OTA treated control group, the final DMSO concentration was 0.1%. In our previous studies, exposure to 200 nM OTA induced kidney and liver toxicity [30,35]. Therefore, in this study, we used 200 nM OTA in HK-2 cells for 48 h to identify the mechanism of toxicity.

### 4.4. Cytotoxicity Assay

HK-2 cells were seeded at 1 × 10^5^ cells/well in a 24 well polystyrene plate (Falcon, Corning, NY, USA). Cell viability was confirmed using the MTT assay [70]. After 24 h of seeding, the cells were treated with OTA for 48 h. After that, the treated medium was removed, 200 μL of MTT solution (1 mg/mL) was added, and incubated for 4 h. Then, 100 μL of DMSO was added after removing the MTT solution to dissolve the formazan crystals formed in living cells. Then, the DMSO with dissolved formazan was transferred to a 96-well plate at 50 μL per well, and its absorbance at 540 nm was measured using a multiplate reader (EL-808, BioTek, Winooski, VT, USA). Three independent experiments (*n* = 3) with three replicate wells were performed.

### 4.5. Total RNA Isolation and Quantitative Real-Time PCR (qRT-PCR)

To examine mRNA expression, the treated HK-2 cells were washed twice with ice-cold PBS, and RNAiso Plus (Takara, Kusatu, Japan) was added to isolate the total RNA. The total RNA concentration in each sample was determined by a Nanodrop 1000 spectrophotometer (Thermo Fisher Scientific, Waltham, MA, USA). Only samples with a Nanodrop A260:280 ratio between 1.8 and 2.1 were used in the experiment. cDNA was synthesized using 2 μg of total RNA and the cocktail solution according to the manufacturer’s instruction in the cDNA synthesis kit (Legene Biosciences, San Diego, CA, USA). In order to confirm the cDNA quality, β-actin, a house keeping gene, was confirmed in agarose gel, and it was also confirmed that the Ct value appeared at a level of 23 to 25 in qRT-PCR. The sequences of primers used in this experiment are shown in Appendix A. Primers were checked by using the NCBI tool Primer-Blast (https://www.ncbi.nlm.nih.gov/tools/primer-blast/, accessed on 10 January 2021). qRT-PCR was performed using 0.5 μL of cDNA, 9.5 μL of primer cocktail, and 10 μL of SYBR green in a total reaction volume of 20 μL on the BioRad iQ5 thermal cycler according to the manufacturer’s protocols (iQ SYBR Green Supermix, Bio-Rad, Hercules, CA, USA). The results were analyzed using the comparative Ct method as described previously [35]. The comparative Ct method was used for relative quantification and normalized using a housekeeping gene (β-actin) and expressed as 2^−△△Ct^ values.

### 4.6. Isolation of Total Cell Lysate and Western Blot Analysis

To isolate total cell lysate, HK-2 cells were lysed using RIPA buffer (25 mM Tris-Cl pH 7.4, 1% Triton X-100, 0.1% SDS, 0.5% deoxycholic acid, 10% glycerol, 150 mM NaCl, 5 mM EDTA, 1 mM PMSF, 5 μg/mL aprotinin, leupeptin and phosphatase inhibitor). The lysed cells were incubated on ice for 30 min and centrifuged at 13,000 rpm and 4 °C for 20 min. The resulting supernatant was used as a total cell lysate. Proteins were separated on 7.5–15% SDS-polyacrylamide gels and electrotransferred to PVDF membranes. The transferred membrane was blocked using 5% skim milk, and immunoblotting was performed using monoclonal AhR (1:200), Smad2/3 (1:1000), p-Smad2/3 (1:1000), HIF-1α (1:200), E-cad (1:1000), FN (1:200), caspase-3 (1:200), Bax (1:200), KIM-1 (1:1000), and GAPDH (1:4000), and then incubated with peroxidase-conjugated secondary antibodies (1:4000). Protein bands were detected using SuperSignal^TM^ West Femto. The intensity of the bands was quantified using Image J program (National Institutes of Health, Maryland, USA), and the protein expressions were normalized to the levels of GAPDH. The control group was set to 1 and the other group was compared with the control group.

### 4.7. Transfection with Small Interfering RNA (siRNA)

HK-2 cells (2 × 10^5^ cells/well in 6 well plate) were transfected with AhR, Smad2/3 and HIF-1α-specific siRNA using Lipofectamine^TM^ RNAiMAX transfection reagent (Invitrogen, Carlsbad, CA, USA) by the reverse transfection method as per the manufacturer’s protocol. Briefly, after transfection with 100 pmol siRNA in 500 μL of Opti-MEM and 5 uL Lipofectamine^TM^ RNAiMAX for 48 h, cells were treated with 200 nM OTA for 48 h and then isolated for experimental purposes such as qRT-PCR or Western blot. The transfection was repeated 3 times for each siRNA.

### 4.8. RNA-sequencing

The quality of RNA used for RNA-sequencing was evaluated by Agilent 2100 bioanalyzer using the RNA 6000 Nano Chip (Agilent Technologies, Amstelveen, Netherlands), and RNA quantification was performed using ND-2000 Spectrophotometer (Thermo Inc., DE, USA).

An Ion AmpliSeq^TM^ Transcriptome library was constructed with the Ion Transcriptome Human Gene Expression Kit (Thermo Fisher Scientific, Waltham, MA, USA) as manufacture’s instruction, and as published [71]. 50 ng of total RNA were reverse transcribed to make cDNA by random priming. cDNA product was amplified target genes using the Ion AmpliSeq Transcriptome Human Gene Expression Core Panel with the Ion AmpliSeq^TM^ Library Kit which is designed for the targeted amplification of more than 20,000 human RefSeq genes simultaneously in a single primer pool. Short amplicons (~100 base pairs (bp)) for the target genes are amplified. After primer digestion, adapters and molecular barcodes were ligated to the amplicons followed by magnetic bead purification. This library concentration were measured using Ion Library Quantitation Kit (Thermo Fisher Scientific, Walthan, MA, USA) according to the manufacturer’s recommendation. Multiple libraries were multiplexed and clonally amplified using the Ion Chef System, and were sequenced on the Ion Torrent S5XL machine (Thermo Fisher Scientific, Waltham, MA, USA).

### 4.9. RNA-seq Data Analysis

All sequencing data was processed on Ion S5xl Sequencer (Thermo Fisher Scientific, Waltham, MA, USA) and transferred to the Ion Proton™ Torrent Server for primary data analysis with gene-level transcript quantification from sequence read data performed using AmpliSeq RNA Plug in (ver 5.6.0.3) by Torrent Suite Software (Thermo Fisher Scientific, Waltham, MA, USA). Identification of up or down-regulated genes was performed using the Excel-based Differentially Expressed Gene Analysis software (ExDEGA version 1.6.3, e-Biogen, Seoul, Korea). The DEG list was filtered using a 5% false discovery rate (FDR) cutoff and |Fold change (FC)| > 2.0 for upregulated and downregulated genes. Gene ontology (GO) and Kyoto Encyclopedia of Genes and Genomes (KEGG, Kyoto, Japan) Pathway Enrichment analysis for OTA was executed using the database for annotation, visualization, and integrated discovery (DAVID, Frederick, MA, USA) version 6.8 (https://david.ncifcrf.gov, accessed on 10 January 2021) functional annotation system [72]. The gene set enrichment analysis (GSEA; Broad Institute) software platform (MSigDB version 6.1, Massachusetts, CA, USA) was used to identify different expression pathways from control cells in OTA-treated cells by comparing them with hallmark gene sets representing specific well-defined biological states [73].

### 4.10. Statistical Analysis of Experiments

All experimental values were expressed as the mean ± standard deviation. Statistically significant differences between groups were calculated using Kruskal–Wallis test and non-parametric Mann–Whitney *U*-test for non-normal distributed data. All statistical analyses were performed in IBM SPSS Statistics version 24 (IBM, Armonk, NY, USA). Different letters indicate significant differences at *p* < 0.05. The data are expressed as mean ± S.D. values of three independent experiments (*n* = 3) with three replicates.

## Figures and Tables

**Figure 1 toxins-13-00190-f001:**
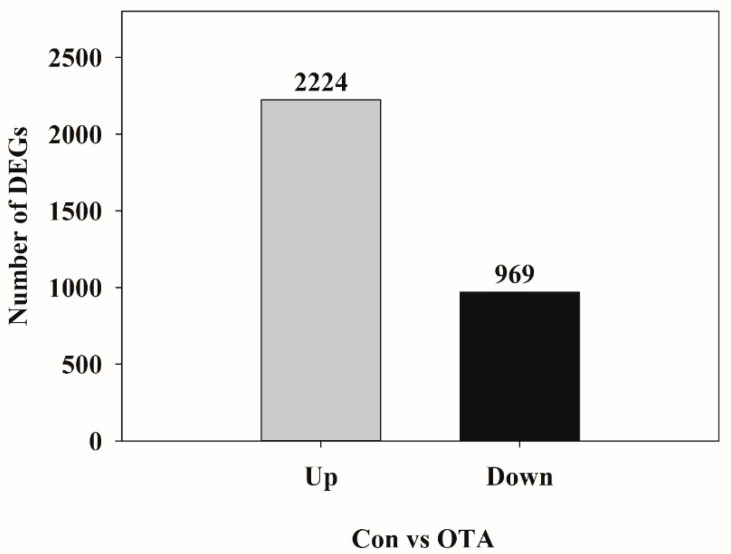
Statistical chart of showing the number of differentially expressed genes (DEGs) in ochratoxin A (OTA)-treated HK-2 cells compared to the control ones. Up represents upregulated DEGs, and Down represents downregulated DEGs. Con, control; OTA, ochratoxin A.

**Figure 2 toxins-13-00190-f002:**
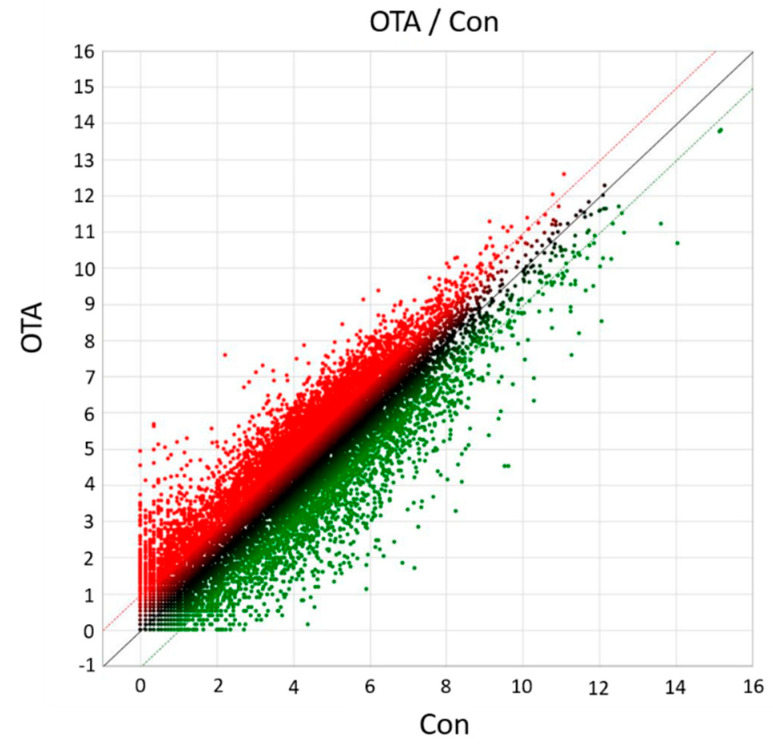
The scatter plot of DEGs between Con and OTA treatment groups. The X-axis indicates the log2 expression values in Con group, and the Y-axis indicates the log2 expression values in OTA group. Each point indicates a particular gene or transcript. Red dots indicate up-regulated genes, green dots indicate down-regulated genes, and black dots indicate genes showing nonsignificant changes.

**Figure 3 toxins-13-00190-f003:**
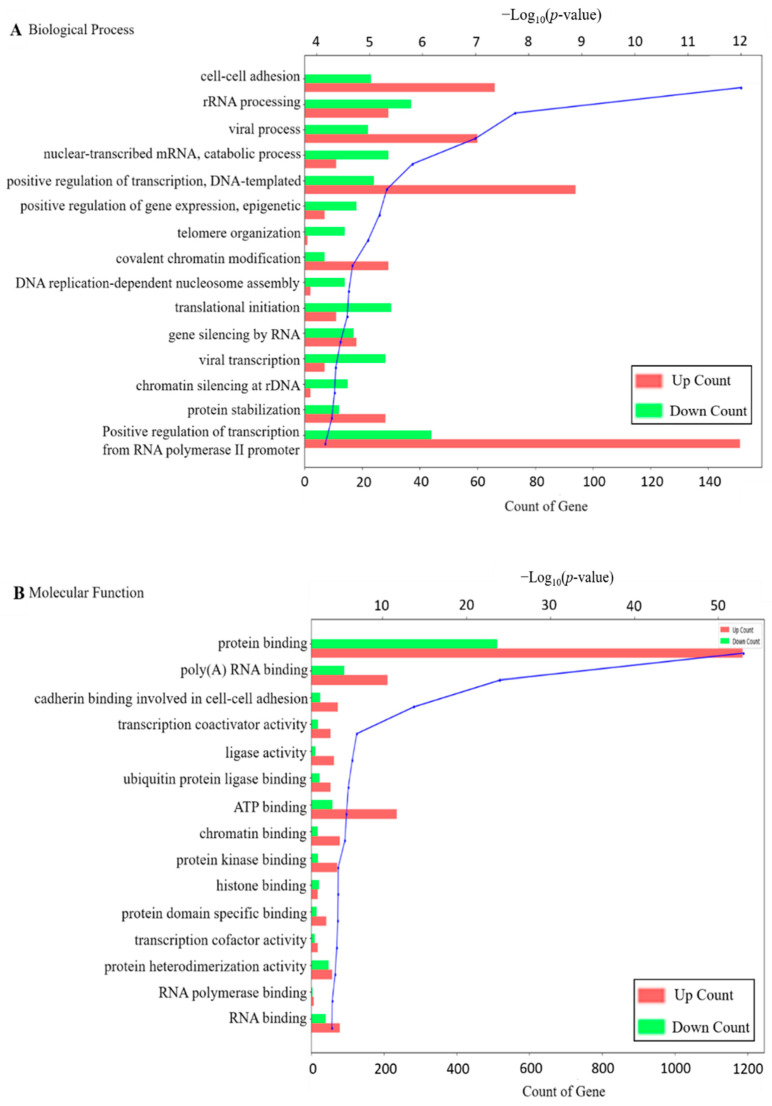
Functional annotation of DEGs. Functional categories of DEGs, broadly separated into (**A**) biological process, (**B**) molecular function and (**C**) cellular component, based on Gene ontology (GO). The blue line of the graphs means −log_10_(*p*-value); (**D**) Top fifteen enriched pathways in HK-2 cells exposed to OTA, analyzed by the KEGG pathway analysis (*p* < 0.05), the size and color of the circle mean −log_10_(*p*-value). The position of the x-axis represents the degree of fold enrichment.

**Figure 4 toxins-13-00190-f004:**
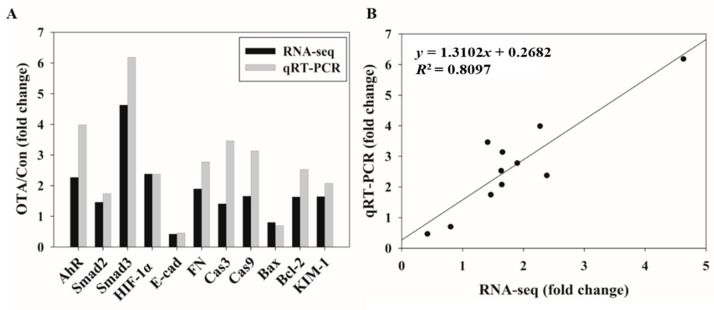
Comparison and validation of OTA-induced gene expression changes determined by RNA-seq and qRT-PCR. (**A**) Fold change of expression value of OTA compared to control measured by RNA-seq and qRT-PCR in eleven selected genes. (**B**) Correlation plot between RNA-seq fold change (FC) compared to qRT-PCR FC. AhR, Aryl Hydrocarbon Receptor; Smad; HIF-1α, Hypoxia-inducible factor 1-alpha; E-cad, epithelial-cadherin; FN, Fibronectin; Cas, Caspase; Bax, Bcl-2-associated X protein; Bcl-2, B-cell lymphoma 2; KIM-1, Kidney injury molecule-1.

**Figure 5 toxins-13-00190-f005:**
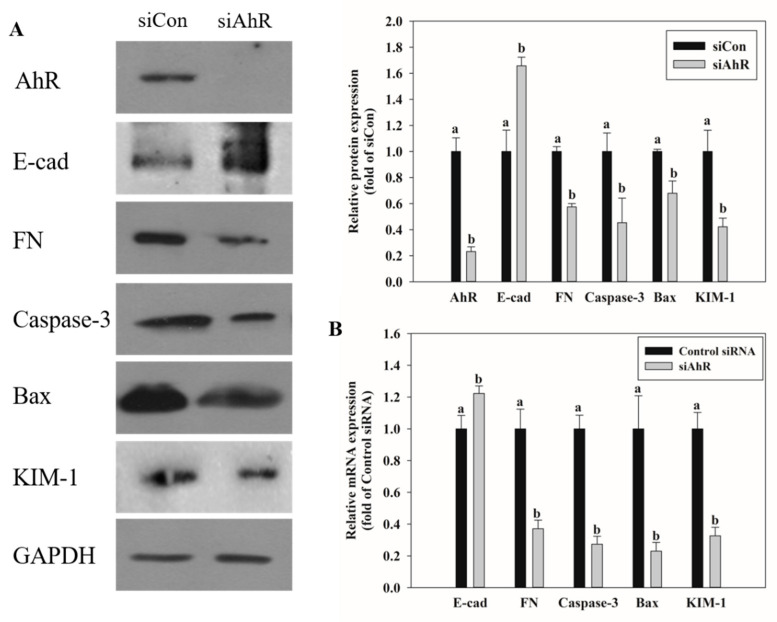
Effects of AhR knockdown on EMT, apoptosis, and kidney injury-related markers expression. (**A**) The protein expression on EMT, apoptosis, and kidney injury-related markers was determined by Western blot after treatment with OTA on HK-2 cells transfected with Control siRNA or siAhR. (**B**) The mRNA expression on EMT, apoptosis, and kidney injury-related markers was determined by qRT-PCR after treatment with OTA on HK-2 cells transfected with scrambled siRNA (Control siRNA) or AhR-specific siRNA (siAhR). The data are expressed as mean ± S.D. values of three independent experiments with three replicate wells. Different letters indicate significant differences compared with control siRNA at *p* < 0.05 by Tukey’s studentized range test.

**Figure 6 toxins-13-00190-f006:**
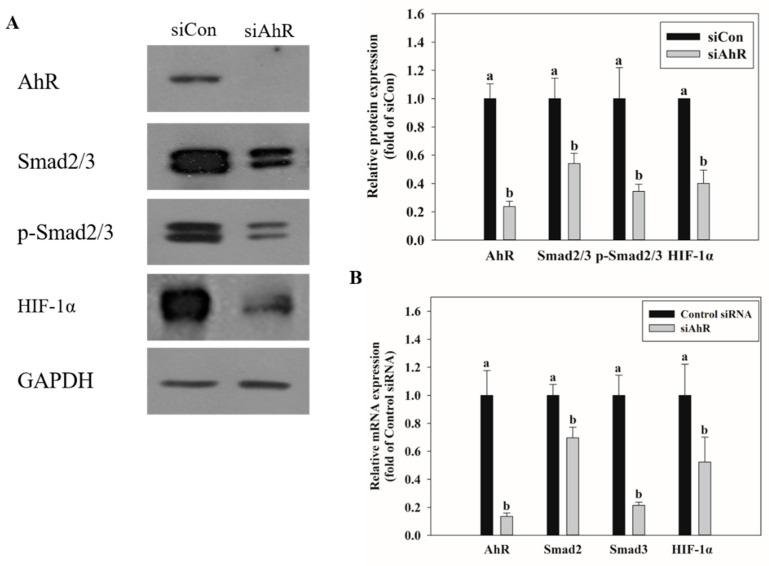
Effects of AhR knockdown on Smad2/3 and HIF-1α expression. (**A**) The protein expression on Smad2/3 and HIF-1α was determined by Western blot after treatment with OTA on HK-2 cells transfected with Control siRNA or siAhR. (**B**) The mRNA expression on Smad2/3 and HIF-1α was determined by qRT-PCR after treatment with OTA on HK-2 cells transfected with Control siRNA or siAhR. The data are expressed as mean± S.D. values of three independent experiments with three replicate wells. Different letters indicate significant differences compared with control siRNA at *p* < 0.05 by Tukey’s studentized range test.

**Figure 7 toxins-13-00190-f007:**
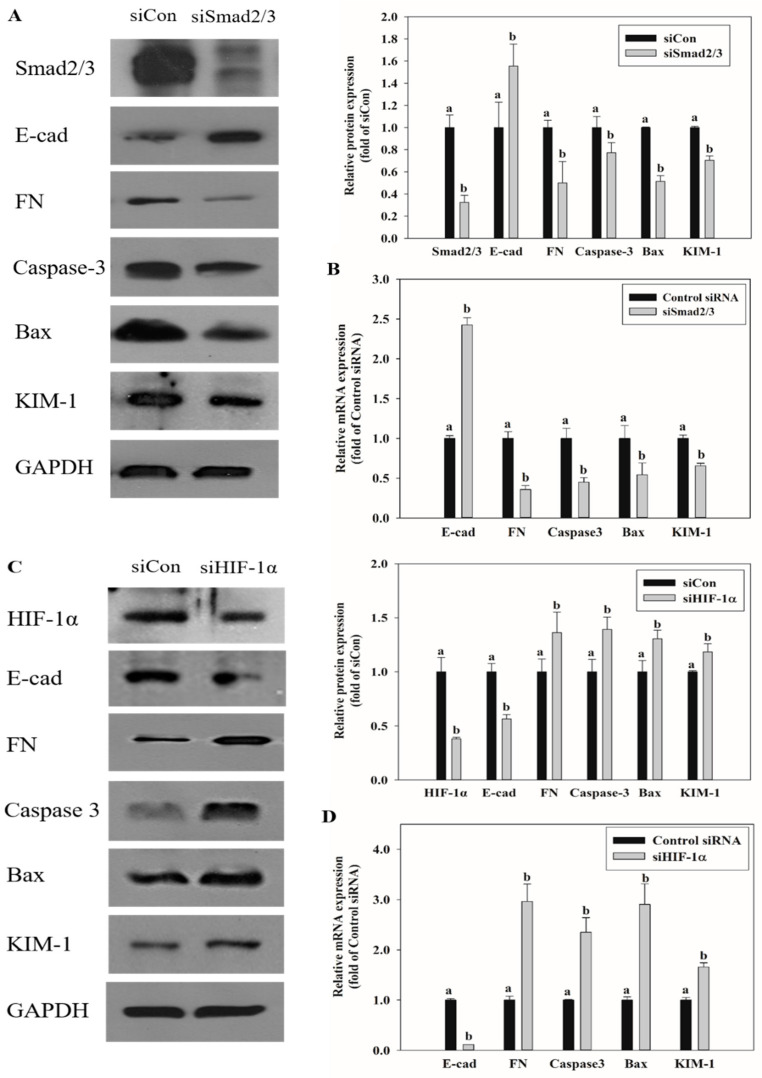
Effects of Smad2/3 and HIF-1α knockdown on EMT, apoptosis, and kidney injury-related markers expression. (**A**) The protein expression on EMT, apoptosis, and kidney injury-related markers was determined by Western blot after treatment with OTA on HK-2 cells transfected with scrambled siRNA (Control siRNA) or Smad2/3-specific siRNA (siSmad2/3). (**B**) The mRNA expression on EMT, apoptosis, and kidney injury-related markers was determined by qRT-PCR after treatment with OTA on HK-2 cells transfected with Control siRNA or siSmad2/3. (**C**) The protein expression on EMT, apoptosis, and kidney injury-related markers was determined by Western blot after treatment with OTA on HK-2 cells transfected with scrambled siRNA (Control siRNA) or HIF-1α -specific siRNA (siHIF-1α). (**D**) The mRNA expression on EMT, apoptosis, and kidney injury-related markers was determined by qRT-PCR after treatment with OTA on HK-2 cells transfected with Control siRNA or siHIF-1α. The data are expressed as mean ± S.D. values of three independent experiments with three replicate wells. Different letters indicate significant differences compared with control siRNA at *p* < 0.05 by Tukey’s studentized range test.

**Table 1 toxins-13-00190-t001:** Gene set enrichment analysis of genes related to OTA.

Gene Set Name	# Genes in Gene Set (K)	# Genes in Overlap (k)	k/K	*p*-Value	FDR q-Value
HYPOXIA	200	57	0.285	1.02 × 10^−27^	5.09 × 10^−26^
TNFA_SIGNALING_VIA_NFKB	200	50	0.250	1.05 × 10^−21^	2.62 × 10^−20^
EPITHELIAL_MESENCHYMAL_TRANSITION	200	46	0.230	1.60 × 10^−18^	2.66 × 10^−17^
GLYCOLYSIS	200	44	0.220	5.26 × 10^−17^	6.58 × 10^−16^
MTORC1_SIGNALING	200	43	0.215	2.89 × 10^−16^	2.89 × 10^−15^
UV_RESPONSE_DN	144	35	0.243	3.19 × 10^−15^	2.65 × 10^−14^
OXIDATIVE_PHOSPHORYLATION	200	40	0.200	4.02 × 10^−14^	2.87 × 10^−13^
UNFOLDED_PROTEIN_RESPONSE	113	29	0.257	1.63 × 10^−13^	1.02 × 10^−12^
ADIPOGENESIS	200	39	0.195	1.96 × 10^−13^	1.09 × 10^−12^
MITOTIC_SPINDLE	199	38	0.191	7.86 × 10^−13^	3.85 × 10^−12^
INFLAMMATORY_RESPONSE	200	38	0.190	9.25 × 10^−13^	3.85 × 10^−12^
P53_PATHWAY	200	38	0.190	9.25 × 10^−13^	3.85 × 10^−12^
INTERFERON_GAMMA_RESPONSE	200	35	0.175	7.99 × 10^−11^	3.07 × 10^−10^
ESTROGEN_RESPONSE_LATE	200	34	0.170	3.30 × 10^−10^	1.18 × 10^−9^
APOPTOSIS	161	30	0.186	3.65 × 10^−10^	1.22 × 10^−9^
TGF_BETA_SIGNALING	54	17	0.315	5.17 × 10^−10^	1.61 × 10^−9^
ESTROGEN_RESPONSE_EARLY	200	30	0.150	6.75 × 10^−8^	1.88 × 10^−7^
IL2_STAT5_SIGNALING	200	30	0.150	6.75 × 10^−8^	1.88 × 10^−7^
COAGULATION	138	24	0.174	8.07 × 10^−8^	2.12 × 10^−7^
XENOBIOTIC_METABOLISM	200	32	0.160	1.86 × 10^−8^	4.05 × 10^−8^

## Data Availability

The data presented in this study are available in article or Appendix A here.

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
