# Peer review of "Transcriptome Analysis Reveals the AhR, Smad2/3, and HIF-1α Pathways as the Mechanism of Ochratoxin A Toxicity in Kidney Cells"

_toxins, 2021, doi:10.3390/toxins13030190_

Round 1

Reviewer 1 Report

Review TOXINS 1095430 „Transcriptome analysis reveals the AhR, Smad2/3, and HIF-1α pathways as the mechansims of ochratoxin A toxicity in kidney cells”

The authors investigated the impact of OTA on human kidney cells (HK-2 cells) as the kidney appears to be one of the main targets of this mycotoxin after ingestion and absorption. Due to the continuous rise of mycotoxin contamination in cereals and other food over the past years in the changing climatic conditions, the topic is highly relevant.

The authors used a variety of techniques to gain more insight into the topic. However, there is some crucial information missing with respective to the experimental setup. As without this information it’s difficult to interpret the data it’s crucial to remedy this. I recommend a thorough revision of the manuscript and have included my points below.

Introduction

Line 33: “through the jejunum with a bioavailability “

Line 64: The abbreviation EMT isn‘t introduced yet, so please use the full name and introduce the abbreviation for further use in the text.

Lines 65-70: These three sentences all have the same message, so please use one concise sentence only.

Material and Methods

In general there is lack of information regarding the actual experimental setup with HK-2 cells. You need to clarify how many replicates were used for the treatment with OTA and how many for the control and of course how many experiments as such. Usually I would expect at least three independent experiments with at least 3-6 replicate wells, thus obtaining at least an N=3.

In particular the control is not explained at all. Is it a DMSO-control and how did you conduct the control in general? As you use all presented data relative to control this is a crucial point and has to be transparently explained in this section!

Please indicate the HK-2 setup for each assay or if you used one experimental setup for a variety of assays, please indicate this as well.

Line 102: You used 24-well plates of which manufacturer? Were they coated? You started OTA treatment 24h after seeding – how did you make sure that HK-2 cells were differentiated?

Statistical analysis: you need to indicate your actual N used for the ANOVA. If you did run your statistics I suspect you tested also the Gaussian distribution of your data in order to decide whether to use a parametric or non-parametric approach. In particular the mRNA expressions (fold of control) need this examination of the Gaussian nature of your data set.

Results

The issue of the above mentioned lack of information continues in the graphs and their legends: are all data presented as actual means ± standard deviation? Then please include this information in the figure legends.

Fig. 2 Font size of your axis is too small to be legible! Please adjust.

Fig.3 Font size of your x-axis and legends (in particular D) is too small to be legible. Please adjust. In A-C there is always a blue line include in the graph, but isn’t explained or defined. Please adjust. Please edit the legend to 3D, for instance “The size and color of the dots expresses –log10 and the position of the x-axis represents the degree of fold enrichment”

Line: If you introduce “EMT” as abbreviation already in the introduction (please see above), you can continue to use the abbreviation without the explanatory terms.

Table 1. Please adjust the first column of your table accordingly to your gene set names so you won’t have any accidental breaks in the table.

Fig. 4: 2 Font size of both axes is too small to be legible! Please adjust.

Fig. 5B: Please include also the western blot for AhR as control for your successful knockdown setup. It’s included only in Fig.6

Fig. 7B: Please include also the western blot for Smad 2/3!

Fig. 7D: Please include also the western blot for HIF-1α!

Discussion

Lines 343-353: You should elaborate more on the relationship between your data (HIF-1α and E-cad etc.), in particular those obtained in the siRNA experiments and how they might explain the mechanisms on kidney cells. Currently you just list what you have done without transferring a meaning to your obtained data. They need to be placed in context!

Author Response

Reviewer 1:

The authors investigated the impact of OTA on human kidney cells (HK-2 cells) as the kidney appears to be one of the main targets of this mycotoxin after ingestion and absorption. Due to the continuous rise of mycotoxin contamination in cereals and other food over the past years in the changing climatic conditions, the topic is highly relevant.

The authors used a variety of techniques to gain more insight into the topic. However, there is some crucial information missing with respective to the experimental setup. As without this information it’s difficult to interpret the data it’s crucial to remedy this. I recommend a thorough revision of the manuscript and have included my points below.

Comments:

  • Line 33: “through the jejunum with a bioavailability “

Þ Answer:

First of all, we’d like to express our gratitude to the reviewer for the careful and critical reading of our manuscript. With highlighted in green color text, we made the correction in response.

We revised as commented in page (P) 1 lines (L)36 as follows:

Ingested OTA is rapidly absorbed through the jejunum with a bioavailability of 97% [4].

  • Line 64: The abbreviation EMT isn‘t introduced yet, so please use the full name and introduce the abbreviation for further use in the text.

Þ Answer:

We added the full name to the part in P2 L70 as follows:

In this research, OTA-induced renal toxicity such as epithelial-mesenchymal transformation (EMT), apoptosis and hypoxia were investigated in human proximal tubular epithelial HK-2 cells.  

  • Lines 65-70: These three sentences all have the same message, so please use one concise sentence only.

Þ Answer:

We made the correction in P2 L69-71 as follows:

In this research, OTA-induced renal toxicity such as epithelial-mesenchymal transformation (EMT), apoptosis and hypoxia were investigated in human proximal tubular epithelial HK-2 cells.

  • In general there is lack of information regarding the actual experimental setup with HK-2 cells. You need to clarify how many replicates were used for the treatment with OTA and how many for the control and of course how many experiments as such. Usually I would expect at least three independent experiments with at least 3-6 replicate wells, thus obtaining at least an N=3.

Þ Answer:

We performed 3 technical replicates for each experiment and the correction in P7 L153-154, P8 L168-169, P9 L194-195, P12 L315-316, P13 L392-395 as follows:

In Figs 5, 6 and 7.

The data are expressed as mean± S.D. values of three independent experiments with three replicate wells.

In P12 L315.

Three independent experiments (n=3) with three replicate wells were performed.

In 392-395.

The data are expressed as mean± S.D. values of three independent experiments (n=3) with three replicates.

  • In particular the control is not explained at all. Is it a DMSO-control and how did you conduct the control in general? As you use all presented data relative to control this is a crucial point and has to be transparently explained in this section!

Þ Answer:

We added information about the groups that were used in the experiment as indicated (presence and concentration of DMSO) in L300-304 as follows:

OTA (purity: 99%, Cfm Oskar Tropitzsch GmbH, Marktredwitz, Germany) was dissolved in DMSO at a concentration of 200 μM and stored at −20°C until used in experiments. In the experiments, to prepare 200 nM OTA, it was diluted into RPMI1640 media. In both the OTA treated group and the non-OTA treated control group, the final DMSO concentration was 0.1%. In our previous studies, exposure to 200 nM OTA induced kidney and liver toxicity [30,35]. Therefore, in this study, we used 200 nM OTA in HK-2 cells for 48 h to identify the mechanism of toxicity.

  • Line 102: You used 24-well plates of which manufacturer? Were they coated? You started OTA treatment 24h after seeding – how did you make sure that HK-2 cells were differentiated?

Þ Answer:

We have added information for the 24 well plate in P12 L308-309 as follows:

HK-2 cells were seeded at 1 × 105 cells/well in a 24 well polystyrene plate (Falcon, Corning, NY, USA). Cell viability was confirmed using the MTT assay [70].

  • Statistical analysis: you need to indicate your actual N used for the ANOVA. If you did run your statistics I suspect you tested also the Gaussian distribution of your data in order to decide whether to use a parametric or non-parametric approach. In particular the mRNA expressions (fold of control) need this examination of the Gaussian nature of your data set.

Þ Answer:

As the reviewer told us, we performed the Shapiro-Wilk`s test to make sure the data follow a normal distribution. As a results, all of data followed a normal distribution. We made  corrections in P13 L387-389 as follows:

4.10. Statistical analysis of experiments

All experimental values were expressed as the mean ± standard deviation. All values to be compared were analyzed for normality using the Shapiro-Wilk test and homogeneity of variances using Levene`s test. One-way analysis of variance (ANOVA) followed by Tukey's studentized range test for the experiments were used to determine all statistically significant differences. SAS version 9.4 (SAS Institute, Cary, NC, USA) was used to perform all tests. Different letters indicate significant differences at p < 0.05. The data are expressed as mean± S.D. values of three independent experiments (n=3) with three replicates.

  • 2 Font size of your axis is too small to be legible! Please adjust.

Þ Answer:

We have modified the figure in P3.

  • 3 Font size of your x-axis and legends (in particular D) is too small to be legible. Please adjust. In A-C there is always a blue line include in the graph, but isn’t explained or defined. Please adjust. Please edit the legend to 3D, for instance “The size and color of the dots expresses –log10 and the position of the x-axis represents the degree of fold enrichment”

Þ Answer:

 We have modified the Figure 3 and its legend in P4-5.

In P5 L 109-112.

Figure 3. Functional annotation of DEGs. Functional categories of DEGs, broadly separated into (A) biological process, (B) molecular function and (C) cellular component, based on Gene ontology (GO). The blue line of the graphs means -log10 (p-value); (D) Top fifteen enriched pathways in HK-2 cells exposed to OTA, analyzed by the KEGG pathway analysis (p<0.05), the size and color of the circle mean –log10 (p-value). The position of the x-axis represents the degree of fold enrichment.

  • Table 1. Please adjust the first column of your table accordingly to your gene set names so you won’t have any accidental breaks in the table.

Þ Answer:

We have modified the Table 1 in P6 L120.

  • 4: 2 Font size of both axes is too small to be legible! Please adjust.

Þ Answer:

 We have modified the Figure 4 in P6.

  • 5B: Please include also the western blot for AhR as control for your successful knockdown setup. It’s included only in Fig.6

Þ Answer:

We added western blot result for AhR in Figure 5 in P7.

  • 7B: Please include also the western blot for Smad 2/3!

Þ Answer:

Answer: We added western blot result for Smad2/3 in Figure 7B in P9.

  • 7D: Please include also the western blot for HIF-1α!

Þ Answer:

We added western blot result for HIF-1α in Figure 7D in P9.

  • Lines 343-353: You should elaborate more on the relationship between your data (HIF-1α and E-cad etc.), in particular those obtained in the siRNA experiments and how they might explain the mechanisms on kidney cells. Currently you just list what you have done without transferring a meaning to your obtained data. They need to be placed in context!

Þ Answer:

We mentioned our findings based on references on the relationship between Ischemia-reperfusion injury and HIF-1a, EMT, and fibrosis in P11 L259-262.

It is known that EMT and fibrosis progress after renal ischemia-reperfusion injury (IRI) [57, 68]. HIF-1α plays a protective role against IRI in the kidney [67]. It protects against IRI by decreasing renal IRI-induced expression of fibrosis and alpha-SMA through increasing HIF- α alpha levels [69]. 

Finally, we really appreciate the reviews’ critical and valuable comments which have been guidance in revising our manuscript. We believe that our manuscript has been much improved due to the revision based on the reviewer’s suggestions.

Reviewer 2 Report

In this work the authors detected through the use of transcriptome and RNA-seq analysis the presence of differentially expressed genes (DEGs) in Ochratoxin A (OTA)-treated HK-2 cells compared to control ones, focusing on the study of hypoxia, apoptosis and the epithelial-mesenchymal transition. The authors demonstrated the involvement of the AhR pathway, Smad2-3,HIF-1 in OTA-induced toxicity by using western Blot and qPCR.

However, although the authors have shown the images of western blot for the analyzed proteins, they should also show histograms with mean and standard deviation obtained by western blot analyses.

Moreover, the authors have explained in “Materials and methods” that they dissolved OTA in DMSO, but it is not clear if they analyzed also cells treated with DMSO alone. This fact would be important to relate the effect were really caused by OTA.

Some modifications needed are indicated below.

Introduction:

  • In line 36, replace the word “renotoxicity” with the word “nephrotoxicity”;
  • Line 39-40, for the phrase “OTA has various possible targets such as the liver, immune system, and brain; however, owing to the function, its main target is the kidneys.” the authors are invited to put some references (Damiano et al. J Cell Physiol. 2020 Jun;235(6):5386-5393. doi: 10.1002/jcp.29425 ;Damiano S et al. 2021 Jan 17;10(1):E125. doi: 10.3390/antiox10010125).

Results:

  • In line 173-175, the authors show the Figure 1 which indicates the number of differentially expressed genes (DEGs) in OTA-treated cells compared to the control ones. However, while in the text ( from line 167 to line 169), the authors have described  this in a very clear way, the caption of the figure does not indicate that the differently expressed genes shown in the Figure 1 are related to OTA-treated cells compared to control cells. In line 175 the authors wrote “Statistical chart of showing the number of differentially expressed genes (DEGs).” and they could add “… in OTA-treated HK-2 cells compared to the control ones” to be clearer.
  • In line 202, Don’t begin a phrase with “And”.
  • Lines 207-208: Since the authors have already used the abbreviation “EMT”, authors can directly use the abbreviation.
  • From Line 213 to line 215, the authors explain they selected a total of 11 genes. The authors could indicate the names of these 11 genes together with their abbreviations.
  • Line 235, in Figure 5 letter B, The authors show only images related to protein bands obtained by western Blot analysis. However, it is also recommended to introduce histograms showing the levels of analyzed proteins.
  • Line 250, in Figure 6 letter B, The authors show only images related to protein bands obtained by western Blot analysis. However, it is also recommended to introduce histograms showing the levels of analyzed proteins.
  • Line 269 Figure 7: Regarding Western Blot, the authors show only images related to protein bands. However, it is also recommended to introduce histograms showing the levels of analyzed proteins.

Discussions:

  • Line 336, the phrase “Mn, [56,57].” is uncompleted

Author Response

Reviewer 2:

In this work the authors detected through the use of transcriptome and RNA-seq analysis the presence of differentially expressed genes (DEGs) in Ochratoxin A (OTA)-treated HK-2 cells compared to control ones, focusing on the study of hypoxia, apoptosis and the epithelial-mesenchymal transition. The authors demonstrated the involvement of the AhR pathway, Smad2-3,HIF-1 in OTA-induced toxicity by using western Blot and qPCR.

However, although the authors have shown the images of western blot for the analyzed proteins, they should also show histograms with mean and standard deviation obtained by western blot analyses.

Moreover, the authors have explained in “Materials and methods” that they dissolved OTA in DMSO, but it is not clear if they analyzed also cells treated with DMSO alone. This fact would be important to relate the effect were really caused by OTA.

Þ Answer:

First of all, we’d like to express our gratitude to the reviewer for the careful and critical reading of our manuscript. With highlighted in green color text, we made the correction in response.

We wrote in detail about the presence and concentration of DMSO in the section on how to treat OTA in page (P) 12 lines (L)300-306 as follows:

OTA (purity: 99%, Cfm Oskar Tropitzsch GmbH, Marktredwitz, Germany) was dissolved in DMSO at a concentration of 200 μM and stored at −20°C until used in experiments. In the experiments, to prepare 200 nM OTA, it was diluted into RPMI1640 media. In both the OTA treated group and the non-OTA treated control group, the final DMSO concentration was 0.1%. In our previous studies, exposure to 200 nM OTA induced kidney and liver toxicity [30,35]. Therefore, in this study, we used 200 nM OTA in HK-2 cells for 48 h to identify the mechanism of toxicity.

Comments:

  • In line 36, replace the word “renotoxicity” with the word “nephrotoxicity”;

Þ Answer:

We appreciate the reviewer’s comment. We have corrected that part in P1 L39-40 as follows:

The nephrotoxicity, hepatotoxicity, immunotoxicity, neurotoxicity, and genotoxicity of OTA have been reported in various animal species [7-9].

  • Line 39-40, for the phrase “OTA has various possible targets such as the liver, immune system, and brain; however, owing to the function, its main target is the kidneys.” the authors are invited to put some references (Damiano et al. J Cell Physiol. 2020 Jun;235(6):5386-5393. doi: 10.1002/jcp.29425 ;Damiano S et al. 2021 Jan 17;10(1):E125. doi: 10.3390/antiox10010125).

Þ Answer:

We added the references as commented in P1 L43, P14 L419-422 as follows:

OTA has various possible targets such as the liver, immune system, heart and brain; however, owing to the function of the organ, its main target is the kidneys [10-12].

  1. Damiano, S.; Iovane, V.; Squillacioti, C.; Mirabella, N.; Prisco, F.; Ariano, A.; Amenta, M.; Giordano, A.; Florio, S.; Ciarcia, R. Red orange and lemon extract prevents the renal toxicity induced by ochratoxin A in rats. Journal of cellular physiology 2020, 235, 5386-5393.
  2. Damiano, S.; Longobardi, C.; Andretta, E.; Prisco, F.; Piegari, G.; Squillacioti, C.; Montagnaro, S.; Pagnini, F.; Badino, P.; Florio, S. Antioxidative Effects of Curcumin on the Hepatotoxicity Induced by Ochratoxin A in Rats. Antioxidants 2021, 10, 125.

  • In line 173-175, the authors show the Figure 1 which indicates the number of differentially expressed genes (DEGs) in OTA-treated cells compared to the control ones. However, while in the text ( from line 167 to line 169), the authors have described  this in a very clear way, the caption of the figure does not indicate that the differently expressed genes shown in the Figure 1 are related to OTA-treated cells compared to control cells. In line 175 the authors wrote “Statistical chart of showing the number of differentially expressed genes (DEGs).” and they could add “… in OTA-treated HK-2 cells compared to the control ones” to be clearer.

Þ Answer:

 We revised legend as you commented in P3 L83-84 as follows:

Figure 1. Statistical chart of showing the number of differentially expressed genes (DEGs) in ochratoxin A (OTA)-treated HK-2 cells compared to the control ones. Up represents upregulated DEGs, and Down represents downregulated DEGs. Con, control; OTA, ochratoxin A.

  • In line 202, Don’t begin a phrase with “And”.

Þ Answer:

We appreciate the reviewer’s comment. We revised legned as you commented in P5 L111 as follows:

(D) Top fifteen enriched pathways in HK-2 cells exposed to OTA, analyzed by the KEGG pathway analysis (p<0.05), the size and color of the circle mean –log10 (p-value). The position of the x-axis represents the degree of fold enrichment.

  • Lines 207-208: Since the authors have already used the abbreviation “EMT”, authors can directly use the abbreviation.

Þ Answer:

We have corrected that part in P5 L116 as follows:

Among them, we focused on hypoxia, EMT, apoptosis, and xenobiotic metabolism as pathways for the toxic mechanism of OTA based on the results of KEGG analysis.

  • From Line 213 to line 215, the authors explain they selected a total of 11 genes. The authors could indicate the names of these 11 genes together with their abbreviations.

Þ Answer:

We added abbreviations for the genes used in P6 L133-135 as follows:

AhR, Aryl Hydrocarbon Receptor; Smad, ; HIF-1α, Hypoxia-inducible factor 1-alpha ; E-cad, epithelial-cadherin; FN, Fibronectin; Cas, Caspase; Bax, Bcl-2-associated X protein; Bcl-2, B-cell lymphoma 2; KIM-1, Kidney injury molecule-1.

  • Line 235, in Figure 5 letter B, The authors show only images related to protein bands obtained by western Blot analysis. However, it is also recommended to introduce histograms showing the levels of analyzed proteins.

Þ Answer:

We appreciate the reviewer’s comment. We quantified the protein band and added it as a graph in P7.

  • Line 250, in Figure 6 letter B, The authors show only images related to protein bands obtained by western Blot analysis. However, it is also recommended to introduce histograms showing the levels of analyzed proteins.

Þ Answer:

We appreciate the reviewer’s comment. We quantified the protein band and added it as a graph in P8.

  • Line 269 Figure 7: Regarding Western Blot, the authors show only images related to protein bands. However, it is also recommended to introduce histograms showing the levels of analyzed proteins.

Þ Answer:

We appreciate the reviewer’s comment. We quantified the protein band and added it as a graph in P9, P13 L349-350 as follows:

The intensity of the bands was quantified using Image J program (National Institutes of Health, Maryland, USA), and the protein expressions were normalized to the levels of GAPDH. The control group was set to 1 and the other group was compared with the control group.

  • Line 336, the phrase “Mn, [56,57].” is uncompleted

Answer: It was our mistake. We deleted that part. We are sorry for the confusion.

Finally, we really appreciate the reviews’ critical and valuable comments which have been guidance in revising our manuscript. We believe that our manuscript has been much improved due to the revision based on the reviewer’s suggestions.

Reviewer 3 Report

Review: OTA effects in kidney cells

Exploring the mechanisms underlying OTA toxicity is important. However, the manuscript absolutely needs refinement. A number of suggestions can be found here:

Abstract:

  • Line 18: how do you know that these pathways were further suppressed. This is a hypothesis, but the authors have not evidence that this is really taking place as they assume. Therefore, this statement should be expressed more vaguely.

Introduction:

  • The genus and species names should be shown in italics.
  • Line 32: why re-absorbed? It is absorbed.
  • Line 38: the references no 7 and 8 are not optimal. There are better ones available to show the effects of OTA in different groups of animals, e.g. rodents, ruminants, fish ect.
  • Line 40 to 51: you mean the function of the organ? The you should also mention the heart as a target, since this organ also is in contact with a lot of blood over time. And effects of OTA on the heart have already been reported.
  • Line 52 to 60: the transcriptome shows which genes have been used to produce RNA, but this does not mean that all the RNA has also been used to produce proteins. This should be mentioned as a limitation of RNAseq
  • Line 62: much is known about renal OTA toxicity but the introduction does not mention this in detail. More details should be added.
  • The last section of the introduction should mention the reasons why this study has been conducted. It should not mention the results.

Materials and Methods

  • Section 2.3 does not mention the purity of OTA. And how much DMSO was used in the exposures. Was DMSO also added to controls?
  • Section 2.4. How many wells have been treated for each assay? How much MTT was used for the cytotox assay?
  • Line 111: how much RNA was extracted in total and how was the quality of the RNA checked?
  • Line 112: which primers are used for cDNA synthesis? How much cDNA was obtained and was the quality of the cDNA checked?
  • The PCRs reactions are poorly described. More details are needed here.
  • Line 115: the primers are not described in sufficient detail. Do they meet the required quality needs? How have they been validated?
  • Line 117: a comparative Ct method is not sufficiently described here. Have reference genes been used? If so, which ones? And how?
  • Line 119: the RIPA buffer is not described.
  • Line 127: the analyses here are not described in sufficient detail.
  • Line 133: the method should be described in more detail. How many cells were used for the transfection? How many replicates ect.
  • Quality checks and library prep have not sufficiently been described. UMIs have to be used to avoid false reads. How many reads have been obtained in total? From how many replicates?
  • The statistical analysis is not sufficient. Using an ANOVA is not appropriate. Models are recommended for this. Literature on this in the web is available

Results & Discussion

Since the generation of the data and the statistical analyses are unreliable and incorrect these chapters have not been checked. They should be checked if the methods and data handling are okay.

Author Response

Reviewer 3:

Exploring the mechanisms underlying OTA toxicity is important. However, the manuscript absolutely needs refinement. A number of suggestions can be found here:

Comments:

  • Line 18: how do you know that these pathways were further suppressed. This is a hypothesis, but the authors have not evidence that this is really taking place as they assume. Therefore, this statement should be expressed more vaguely.

Þ Answer:

First of all, we’d like to express our gratitude to the reviewer for the careful and critical reading of our manuscript. With highlighted in green color text, we made the correction in response.

We deleted the ambiguous content and rewrote it in page (P) 1 lines (L)16-19 as follows:

Smad2/3 suppression with siRNA could inhibit fibronetcin, caspase-3, Bax, and KIM-1 expression. Fibronetcin, caspase-3, Bax, and KIM-1 expression could be increased with HIF-1α suppression with siRNA. Taken together, these findings suggest that OTA-mediated kidney toxicity via the AhR-Smad2/3-HIF-1α signaling pathways leads to induction of EMT, apoptosis, and kidney injury.

  • The genus and species names should be shown in italics.

Þ Answer:

We modified it to italics in P1 L 28-29 as follows:

Ochratoxin A (OTA) is a mycotoxin that occurs naturally in fungi such as Aspergillus spp. and Penicillium spp.

  • Line 32: why re-absorbed? It is absorbed.

Þ Answer:

We have corrected that part in P1 L36 as follows:

Ingested OTA is rapidly absorbed through the jejunum with a bioavailability of 97% [4].

  • Line 38: the references no 7 and 8 are not optimal. There are better ones available to show the effects of OTA in different groups of animals, e.g. rodents, ruminants, fish ect.

Þ Answer:

We replaced these references with those that studied the effect of OTA in other species such as rodents in P1 L42-43, P14 L413-416 as follows:

P1 L42-43

OTA has various possible targets such as the liver, immune system, heart and brain; however, owing to the function of the organ, its main target is the kidneys [10-12].

P14 L413-416

  1. Pfohl‐Leszkowicz, A.; Manderville, R.A. Ochratoxin A: An overview on toxicity and carcinogenicity in animals and humans. Molecular nutrition & food research 2007, 51, 61-99.
  2. Mally, A.; Dekant, W. Mycotoxins and the kidney: modes of action for renal tumor formation by ochratoxin A in rodents. Molecular nutrition & food research 2009, 53, 467-478.

  • Line 40 to 51: you mean the function of the organ? you should also mention the heart as a target, since this organ also is in contact with a lot of blood over time. And effects of OTA on the heart have already been reported.

Þ Answer:

We appreciate the reviewer’s comment. We added heart to the target of the OTA, and as you commented, we added content to prevent confusion in P1 L42-43, P14 L413-416 as follows:

P1 L42-43

OTA has various possible targets such as the liver, immune system, heart and brain; however, owing to the function of the organ, its main target is the kidneys [10-12].

  • Line 52 to 60: the transcriptome shows which genes have been used to produce RNA, but this does not mean that all the RNA has also been used to produce proteins. This should be mentioned as a limitation of RNAseq

Þ Answer:

As you mentioned, we added a sentence about a limitation of RNAseq in P2 L56-57 as follows:

Although there is a limitation of RNA-seq that not all RNAs have been used to produce proteins, the use of RNA-seq technology to analyze differentially expressed genes (DEGs) is a reliable method for understanding the interactions between specific molecules and for conducting research on the mechanisms that cause toxicity [20,21].

  • Line 62: much is known about renal OTA toxicity but the introduction does not mention this in detail. More details should be added.

Þ Answer:

We have added specific symptoms regarding the renal toxicity of OTA in P2 L66-67 as follows:

Although much is known about the renal toxicity of OTA such as inflammation, apoptosis and pyroptosis [26,27], studies on the mechanisms underlying renal toxicity are still insufficient.

  • The last section of the introduction should mention the reasons why this study has been conducted. It should not mention the results.

Þ Answer:

We deleted the pointed out because it was not appropriate to enter the introduction part in P2 L69-71 as follows:

In this research, OTA-induced renal toxicity such as epithelial-mesenchymal transformation (EMT), apoptosis and hypoxia were investigated in human proximal tubular epithelial HK-2 cells.

  • Section 2.3 does not mention the purity of OTA. And how much DMSO was used in the exposures. Was DMSO also added to controls?

Þ Answer:

We added about the purity of the OTA, and in the experiment we added the concentration of DMSO treated per group to the cells in P12 L300-304 as follows:

OTA (purity: 99%, Cfm Oskar Tropitzsch GmbH, Marktredwitz, Germany) was dissolved in DMSO at a concentration of 200 μM and stored at −20°C until used in experiments. In the experiments, to prepare 200 nM OTA, it was diluted into RPMI1640 media. In both the OTA treated group and the non-OTA treated control group, the final DMSO concentration was 0.1%.

  • Section 2.4. How many wells have been treated for each assay? How much MTT was used for the cytotox assay?

Þ Answer:

We have added the number of repetitions for the experiment. Also, the concentration of MTT solution used in the MTT assay was also added in P12, L311, L315-316 as follows:

After that, the treated medium was removed, 200 μL of MTT solution (1 mg/mL) was added, and incubated for 4 h. Then, 100 μL of DMSO was added after removing the MTT solution to dissolve the formazan crystals formed in living cells. Then, the DMSO with dissolved formazan was transferred to a 96-well plate at 50 μL per well, and its absorbance at 540 nm was measured using a multiplate reader (EL-808, BioTek, Winooski, VT, USA). Three independent experiments (n=3) with three replicate wells were performed.

  • Line 111: how much RNA was extracted in total and how was the quality of the RNA checked?

Þ Answer:

We added the content to the part where the concentration and quality of RNA were checked using Nanodrop in P12 L320-322 as follows:

The total RNA concentration in each sample was determined by a Nanodrop 1000 spectrophotometer (Thermo Fisher Scientific, Waltham, MA, USA). Only samples with a Nanodrop A260:280 ratio between 1.8 and 2.1 were used in the experiment.

  • Line 112: which primers are used for cDNA synthesis? How much cDNA was obtained and was the quality of the cDNA checked?

Þ Answer:

We added methods using agarose gel and real-time PCR to measure the quality of cDNA in P 12 L325-327.

In order to confirm the cDNA quality, β-actin, a house keeping gene, was confirmed in agarose gel, and it was also confirmed that the Ct value appeared at a level of 23 to 25 in qRT-PCR.

  • The PCRs reactions are poorly described. More details are needed here.

Þ Answer:

We added information on the volume of reagents and cDNA required for qRT-PCR in P12 L 329-331 as follows:

qRT-PCR was performed using 0.5 μL of cDNA, 9.5 μL of primer cocktail, and 10 μL of SYBR green in a total reaction volume of 20 μL on the BioRad iQ5 thermal cycler according to the manufacturer`s protocols (iQ SYBR Green Supermix, Bio-Rad, Hercules, CA, USA).

  • Line 115: the primers are not described in sufficient detail. Do they meet the required quality needs? How have they been validated?

Þ Answer:

The sequence of primers used in the experiment was written in the supplement table, and primers were identified using the NCBI tool primer-blast in P12 L327-329 as follows:

Primers were checked by using the NCBI tool Primer-Blast (https://www.ncbi.nlm.nih.gov/tools/primer-blast/).

  • Line 117: a comparative Ct method is not sufficiently described here. Have reference genes been used? If so, which ones? And how?

Þ Answer:

The method was written in more detail, and the content using β-actin as a reference gene was added in P12 L333-335.

The results were analyzed using the comparative Ct method as described previously [35]. The comparative Ct method was used for relative quantification and normalized using a housekeeping gene (β-actin) and expressed as 2-△△Ct values.

  • Line 119: the RIPA buffer is not described.

Þ Answer:

We have written in detail about the RIPA buffer in P12 L337-339 as follows:

To isolate total cell lysate, HK-2 cells were lysed using RIPA buffer (25 mM Tris-Cl pH 7.4, 1% Triton X-100, 0.1% SDS, 0.5% deoxycholic acid, 10% glycerol, 150 mM NaCl, 5 mM EDTA, 1 mM PMSF, 5 μg/mL aprotinin, leupeptin and phosphatase inhibitor).

  • Line 127: the analyses here are not described in sufficient detail.

Þ Answer:

We have written and added details about the analysis in P12 L337-339 as follows:

To isolate total cell lysate, HK-2 cells were lysed using RIPA buffer (25 mM Tris-Cl pH 7.4, 1% Triton X-100, 0.1% SDS, 0.5% deoxycholic acid, 10% glycerol, 150 mM NaCl, 5 mM EDTA, 1 mM PMSF, 5 μg/mL aprotinin, leupeptin and phosphatase inhibitor).

  • Line 133: the method should be described in more detail. How many cells were used for the transfection? How many replicates ect.

Þ Answer:

We appreciate the reviewer’s comment. We added the conditions used in the experiment in P13 L352-358 as follows:

HK-2 cells (2× 105 cells/well in 6 well plate) were transfected with AhR, Smad2/3 and HIF-1α-specific siRNA using LipofectamineTM RNAiMAX transfection reagent (Invitrogen, Carlsbad, CA, USA) by the reverse transfection method as per the manufacturer`s protocol. Briefly, after transfection with 100 pmol siRNA in 500 μL of Opti-MEM and 5 uL LipofectamineTM RNAiMAX for 48 h, cells were treated with 200 nM OTA for 48 h and then isolated for experimental purposes such as qRT-PCR or western blot. The transfection was repeated 3 times for each siRNA.

  • Quality checks and library prep have not sufficiently been described. UMIs have to be used to avoid false reads. How many reads have been obtained in total? From how many replicates?

Þ Answer:

We added the quality check part and added the normalized method using software in P13 L360-363, 372-373 as follows:

The quality of RNA used for RNA-sequencing was evaluated by Agilent 2100 bioanalyzer using the RNA 6000 Nano Chip (Agilent Technologies, Amstelveen, Netherlands), and RNA quantification was performed using ND-2000 Spectrophotometer (Thermo Inc., DE, USA). On the Ion Torrent S5XL platform, AmpliSeq libraries were constructed and sequenced according to the manufacturer`s instructions. For the analysis of human genes in HK-2 cells, the Ion AmpliSeq Transcriptome Human Gene Expression Kit (Thermo Fisher Scientific, Waltham, MA, USA) was used, which is designed for the targeted amplification of more than 20,000 human RefSeq genes simultaneously in a single primer pool. Short amplicons (~100 base pairs (bp)) for the target genes are amplified and 50 ng of total RNA was used to prepare the cDNA library for each group of samples. Multiple libraries were multiplexed and clonally amplified using the Ion Chef System, and were sequenced on the Ion Torrent S5XL machine (Thermo Fisher Scientific, Waltham, MA, USA). Normalized method using ampliSeq RNA Plug in (ver 5.6.0.3) by Torrent Suite Software (Thermo Fisher Scientific, Waltham, MA, USA).

  • The statistical analysis is not sufficient. Using an ANOVA is not appropriate. Models are recommended for this. Literature on this in the web is available

Þ Answer:

As the reviewer told us, we performed the Shapiro-Wilk`s test to make sure the data follow a normal distribution. As a results, all of data followed a normal distribution in P13 L387-389, L392-394 as follows:

All experimental values were expressed as the mean ± standard deviation. All values to be compared were analyzed for normality using the Shapiro-Wilk test and homogeneity of variances using Levene`s test. One-way analysis of variance (ANOVA) followed by Tukey's studentized range test for the experiments were used to determine all statistically significant differences. SAS version 9.4 (SAS Institute, Cary, NC, USA) was used to perform all tests. Different letters indicate significant differences at p < 0.05. The data are expressed as mean± S.D. values of three independent experiments (n=3) with three replicates.

Finally, we really appreciate the reviews’ critical and valuable comments which have been guidance in revising our manuscript. We believe that our manuscript has been much improved due to the revision based on the reviewer’s suggestions.

Round 2

Reviewer 2 Report

The article is ready to be submitted

Author Response

Comments:

The article is ready to be submitted.

Þ Answer:

First of all, we’d like to express our gratitude to the reviewer for the careful and critical reading of our manuscript.

Reviewer 3 Report

Dear authors,

it is not okay to perform ANOVA caluclations with a sample size of n=3. The reason is that you cannot judge normal distribution with n=3. You have to use non-parametric tests.

And with the RNAseq it is even more complicated because you perform thousands of comparisons and this inflated the statistical probability to find a positive result. This is why you should use models for checking differences in the data sets. I recommend reading the publication e.g. of Li and Li 2018 on Modeling and analysis of RNA-seq data: a review from a statistical perspective published in Quant. Biol  doi:10.1007/s40484-018-0144-7

Author Response

Reviewer 3:

Comments:

  • It is not okay to perform ANOVA caluclations with a sample size of n=3. The reason is that you cannot judge normal distribution with n=3. You have to use non-parametric tests.

Þ Answer:

First of all, we’d like to express our gratitude to the reviewer for the careful and critical reading of our manuscript. With highlighted in green color text, we made the correction in response.

We performed Kruskal-Wallis test and Mann-Whitney U-test, which are non-parametric tests, using SPSS for all results. The outcome of the significance of the data using the Kruskal-Wallis test and the Mann-Whitney U-test did not differ from that of the anova test. In page (P) 13-14, lines (L) 395-398, the non-parametric test concerned was added as follows:

4.10. Statistical analysis of experiments

“All experimental values were expressed as the mean ± standard deviation. Statistically significant differences between groups were calculated using Kruskal-Wallis test and non-parametric Mann-Whitney U-test for non-normal distributed data. All statistical analyses were performed in IBM SPSS Statistics version 24. Different letters indicate significant differences at p < 0.05. The data are expressed as mean± S.D. values of three independent experiments (n=3) with three replicates.”

  • And with the RNAseq it is even more complicated because you perform thousands of comparisons and this inflated the statistical probability to find a positive result. This is why you should use models for checking differences in the data sets.

Þ Answer:

Please notice that The RNA-seq we performed by Ion AmpliSeq sequencing was developed by Thermo Fisher Scientific, and the ampliseq transcriptome data was processed according to the manufacturer`s protocol. More information on the Ion AmpliSeq system with reference (71) and raw data analysis have been added in (P) 13, lines (L) 364-383 as follows:

4.8. RNA-sequencing

The quality of RNA used for RNA-sequencing was evaluated by Agilent 2100 bioanalyzer using the RNA 6000 Nano Chip (Agilent Technologies, Amstelveen, Netherlands), and RNA quantification was performed using ND-2000 Spectrophotometer (Thermo Inc., DE, USA).

An Ion AmpliSeqTM Transcriptome library was constructed with the Ion Transcriptome Human Gene Expression Kit (Thermo Fisher Scientific, Waltham, MA, USA) as manufacture`s instruction, and as published [71]. 50 ng of total RNA were reverse transcribed to make cDNA by random priming. cDNA product was amplified target genes using the Ion AmpliSeq Transcriptome Human Gene Expression Core Panel with the Ion AmpliSeqTM Library Kit which is designed for the targeted amplification of more than 20,000 human RefSeq genes simultaneously in a single primer pool. Short amplicons (~100 base pairs (bp)) for the target genes are amplified. After primer digestion, adapters and molecular barcodes were ligated to the amplicons followed by magnetic bead purification. This library concentration were measured using Ion Library Quantitation Kit (Thermo Fisher Scientific, Walthan, MA, USA) according to the manufacturer`s recommendation. Multiple libraries were multiplexed and clonally amplified using the Ion Chef System, and were sequenced on the Ion Torrent S5XL machine (Thermo Fisher Scientific, Waltham, MA, USA).

4.9. RNA-seq data analysis

All sequencing data was processed on Ion S5xl Sequencer (Thermo Fisher Scientific, ) and transferred to the Ion Proton™ Torrent Server for primary data analysis with gene-level transcript quantification from sequence read data performed using AmpliSeq RNA Plug in (ver 5.6.0.3) by Torrent Suite Software (Thermo Fisher Scientific, Waltham, MA, USA).~”

Again, we very appreciate the critical and useful input of the reviewers who have driven us to revise our manuscript. We believe that our manuscript has improved significantly as a result of a revision following the advice of the reviewers.

Round 3

Reviewer 3 Report

no comments.

This manuscript is a resubmission of an earlier submission. The following is a list of the peer review reports and author responses from that submission.